# One Copy Number Variation within the Angiopoietin-1 Gene Is Associated with Leizhou Black Goat Meat Quality

**DOI:** 10.3390/ani14182682

**Published:** 2024-09-14

**Authors:** Qun Wu, Xiaotao Han, Yuelang Zhang, Hu Liu, Hanlin Zhou, Ke Wang, Jiancheng Han

**Affiliations:** 1Zhanjiang Experimental Station, Chinese Academy of Tropical Agricultural Sciences, No. 5, Shetan Road, Xiashan Area, Zhanjiang 524013, China; wuqun.2006@163.com (Q.W.); xthan0521@163.com (X.H.); liuh2018@lzu.edu.cn (H.L.); zhouhanlin8@163.com (H.Z.); 2Hainan Institute, Zhejiang University, Sanya 572024, China; zhangyuelang@zju.edu.cn

**Keywords:** angiopoietin 1, CNV, goats, meat quality

## Abstract

**Simple Summary:**

The *ANGPT1* gene is crucial for angiogenesis and muscle growth. This study analyzed three *ANGPT1* copy number variations (CNVs) in 417 Leizhou black goats using quantitative PCR (qPCR). Specifically, CNV-1 (ARS1_chr14:24950001-24953600), which overlaps with protein-coding regions, showed that a higher copy number (≥3) was significantly associated with increased *ANGPT1* mRNA expression in muscle and improved traits such as carcass weight and muscle quality. These results suggest CNV-1’s gain-of-copies genotype could be an effective marker for enhancing growth and meat quality in targeted breeding programs.

**Abstract:**

The *ANGPT1* gene plays a crucial role in the regulation of angiogenesis and muscle growth, with previous studies identifying copy number variations (CNVs) within this gene among Leizhou black goats. In this study, we investigated three *ANGPT1* CNVs in 417 individuals of LZBG using quantitative PCR (qPCR), examining the impact of different CNV types on the *ANGPT1* gene expression and their associations with growth and meat quality traits. Notably, the *ANGPT1* CNV-1 (ARS1_chr14:24950001-24953600) overlaps with protein-coding regions and conserved domains; its gain-of-copies genotype (copies ≥ 3) was significantly correlated with *ANGPT1* mRNA expression in muscle tissue (*p* < 0.01). Furthermore, the gain-of-copies genotype of CNV-1 demonstrated significant correlations with various phenotypic traits, including carcass weight, body weight, shear stress, chest circumference, and cross-sectional area of longissimus dorsi muscle. These findings indicate that the CNV-1 gain-of-copies genotype in the *ANGPT1* gene may serve as a valuable marker for selecting Leizhou black goats exhibiting enhanced growth and muscular development characteristics, thereby holding potential applications in targeted breeding programs aimed at improving meat quality.

## 1. Introduction

Goats are regarded as the perfect animal model for researching climate change when compared to other domestic species. Owing to their robust thermal and drought resistance, ability to flourish on limited pastures, and higher disease resistance, they are highly valued [1]. Compared to other red meats such as beef or lamb, chevon is lower in fat and cholesterol, offering a tasty and nutritious alternative [2].

The angiopoietin 1 (*ANGPT1*) gene, known for encoding a secretory glycoprotein, plays a crucial role in various functions. It activates receptors through tyrosine phosphorylation, regulating interactions between endothelial cells and their surroundings [3,4]. This gene is pivotal in inhibiting endothelial cell permeability, primarily contributing to angiogenesis and skeletal muscle regeneration [5]. On one hand, the ANGPT1 protein can regulate cell proliferation via the ERK kinase pathway [6], and on the other hand, it can induce cell differentiation by binding to Tie-2 and activating the mTOR signaling pathway [7]. In human skeletal myoblasts cultured in vitro, the addition of recombinant ANGPT1 protein enhanced cell survival, proliferation, migration, and differentiation into myotubes [8].

Copy number variation (CNV) refers to differences in the number of copies of DNA segments, larger than 1 kilobase, among individuals in a population, affecting gene expression and phenotypic traits [9,10]. They can disrupt gene dosage, leading to haploinsufficiency when a gene copy is lost, or gene overexpression when copies are gained, potentially affecting cellular processes and contributing to diseases [11,12,13]. In livestock, CNV analysis can help identify regions associated with traits for selective breeding and genetic improvement [14,15]. Comparing CNV differences between infected and uninfected individuals can also pinpoint regions linked to disease resistance [16].

The Leizhou black goat (LZBG), a predominant breed in China’s tropical regions, often shows signs of amyotrophy and undernourishment in its young [17]. These signs lead to lower productivity and diminished market worth [18]. In our earlier research of the LZBG, we found several CNVs in the *ANGPT1* gene, which may be essential for regulating muscle development. This study aimed to investigate how CNVs within the *ANGPT1* gene might affect meat quality in goats. While the existing research methodologies are inadequate for directly conducting experiments on goat ANGPT1 protein, we hypothesize that CNV-1’s main role might be in regulating ANGPT1 gene expression by impacting its transcriptional structure. These results will enhance comprehension regarding the *ANGPT1* gene’s influence on goat muscle and establish a foundation for enhancing goat genetics.

## 2. Materials and Methods

### 2.1. Collection of Samples and Data

Ear tissues were collected from female LZBGs (*n* = 417, aged 720 ± 30 days) at the Leizhou Black Goat Breeding Farm. The goats were selected by simple randomization using a random sequence generated by the RAND formula in EXCEL (LTSC MSO 2021) from a total of 952 doses of appropriate age from 9 pedigrees of different blood origin, under the same feeding and management conditions as outlined by Gu et al. [19]. Various growth metrics, including chest circumference (CC), body height (BH), body weight (BW), cannon circumference (CAC), and body oblique length (BOL), were measured and recorded. Measurements were repeated three times, and the median value was taken. In total, 80 of the 417 LZBGs were randomly selected for slaughter, and several meat quality parameters (shear stress, SS; the cross-sectional area of the longissimus dorsi muscle, CALM; carcass weight, CW; water holding capacity, WHC; water loss rate, WLR) were evaluated. Additionally, gene expression analysis was performed on randomly sampled tissues (longissimus dorsi muscle, heart, brain, liver, ovary, gluteofemoral biceps, back skin, kidney, and colon) of 12 LZBGs. Expression profiling was also conducted on 15 longissimus dorsi muscle samples from LZBG goats at 30 days, 6 months, 12 months, 2 years, and 4 years of age. All samples were immediately preserved in RNA later and stored at −80 °C. The experimental procedures were authorized by the Review Committee of the Chinese Academy of Tropical Agricultural Sciences and conducted in compliance with the ethics commission guidelines (CATAS-20230019ZES).

### 2.2. Bioinformatics Methods

We sourced the amino acid sequences of the ANGPT1 protein from 10 species using the NCBI database (https://www.ncbi.nlm.nih.gov/protein, accessed on 15 February 2024). For a comprehensive comparison of the similarities and differences in these sequences, we selected the isoform with the longest length and most extensive gene coverage from the following species: *Bos taurus*, *Bubalus bubalis*, *Bos indicus*, *Bos mutus*, *Capra hircus*, *Homo sapiens*, *Mus musculus*, *Rattus rattus*, *Sus scrofa*, and *Ovis aries* [20]. To compare multiple sequences and build a phylogenetic tree, we employed the MUSCLE tool available in MEGA-11.0.13 alongside the Neighbor-Joining (NJ) method. For a detailed exploration of the ANGPT1 protein’s structural attributes and functional motifs, the MEME (https://meme-suite.org/, accessed on 17 February 2024) suite was employed [21]. Examining ANGPT1 protein motifs across different species provided us with valuable insights into the conserved elements and variations in the protein’s motif structure, deepening our comprehension of its biological significance. We utilized the AlphaFold and SOPMA online tools to predict the advanced structure of the protein (https://npsa-prabi.ibcp.fr/, accessed on 17 February 2024) [22]. Furthermore, we examined the conservative domain structure and function of the ANGPT1 protein using CDD NCBI (https://www.ncbi.nlm.nih.gov/cdd/, accessed on 20 February 2024) [23].

### 2.3. Genomic DNA and Total RNA Extraction

The Animal Tissues/Cells Genomic DNA Extraction Kit (Solarbio, Beijing, China) was used to extract genomic DNA, and the concentration was measured by Nanodrop One spectrophotometer (Thermo Fisher, Waltham, MA, USA). Total RNA was isolated using the Trizol method for the qPCR process. Following the manufacturer’s guidelines, the 1st-strand cDNA was prepared with the Reagent Kit (PrimeScript™ RT, Takara, Shiga, Japan).

### 2.4. Primer Design

Based on information from the Goat Pan-genome Database (http://animal.omics.pro/, accessed on 24 February 2024) [24], the Vargoats Database (https://www.goatgenome.org/, accessed on 24 February 2024) [25], and the LZBG whole genome sequencing results (Compared to ARS1.0, which is sourced from NCBI-genome, https://www.ncbi.nlm.nih.gov/datasets/genome, accessed on 24 February 2024) [20], we discovered five CNVs in the goat *ANGPT1* gene, designated as CNV-1 (ARS1_chr14:24950001-24953600), CNV-2 (ARS1_chr14:25038801-25040400), and CNV-3 (ARS1_chr14:25111601-25114400). Additionally, two potential CNVs (ARS1_chr14:25120801-25122800 and ARS1_chr14:25244401-25246000) listed in the databases were not found in our sequencing data. We then designed six primer pairs, with *MC1R* and *GAPDH* serving as reference genes [26], and the specific primers are listed in Appendix A.

### 2.5. Statistical Analysis

The genomic DNA obtained from ear tissues was utilized for copy number assessment through qPCR, following the procedure outlined by Weaver et al. [27]. The reaction conditions and thermal profile were in accordance with the methodology detailed by Wang et al. [28]. The relative copy number (RCN) was computed from the ΔΔCt value using the formula as follows [27]:Relative copy number (RCN) = 2^−ΔΔCt^
Copies per diploid genome = 2 × 2^−ΔΔCt^

The CNVs were categorized into three groups: loss (copy = 1), normal (copies = 2), and gain (copies ≥ 3). *ANGPT1* mRNA expression was examined using cDNA from different tissues. PCR amplification was performed as described by Wang et al. [29]. Relative gene expression was assessed using the 2^−ΔΔCt^ method, with GAPDH serving as the internal control. To examine the association between CNV genotypes and traits, statistical analyses were conducted using SPSS software (version 26), and a one-way analysis of variance (ANOVA) was employed for this purpose. Similarly, we used one-way ANOVA to evaluate the impact of CNV genotypes on mRNA expression levels. The analysis was based on a linear model:Y*ij* = α*i* + e*j* + *μ*
where Y*ij*: the evaluation of traits; α*i*: the fixed factor CNV types; e*j*: the random error; and *μ*: the overall mean [30].

## 3. Results

### 3.1. Analysis of the Conservation and Evolution of ANGPT1

The ANGPT1 protein’s amino acid sequences were compared and examined across 10 different animal species. It was noted that the ANGPT1 protein’s structure was highly conserved in nine species, except for humans (Figure 1A). This discrepancy is also evident in the localization of motifs. In contrast to other species, the human ANGPT1 protein lacks the fibrinogen polymerization pocket motifs (Figure 1C). Analysis of the phylogenetic tree indicated a close relationship among goats, sheep, and other ruminants (Figure 1B). Predictions of the protein structure revealed that the ANGPT1 protein primarily consists of the Mplasa_alph_rch superfamily in the head and Fibrinogen-related domains (FReDs) in the tail (Figure 1D). Since only CNV-1 among the three CNVs is situated in the exon region, the protein structure of the *ANGPT1* CNV-1 region was examined. The findings indicated that the CNV-1 region includes the alpha/beta/gamma chain, fibrinogen, Fibrinogen-like C-terminal, and C-terminal globular domain (Figure 2). The goat *ANGPT1* gene encodes a total of 488 amino acids, with the CNV-1 region comprising 43 amino acids.

### 3.2. ANGPT1 Expression and Its Associations with CNVs

*ANGPT1* expression was assessed in 10 different tissues from the LZBG (Figure 3A), and *ANGPT1* mRNA was widely present in the tissues of adult goats. Specifically, the expression levels of *ANGPT1* mRNA were significantly higher in the heart and longissimus dorsi muscle compared to other tissues among the 10 analyzed (*p* < 0.05). Figure 3B illustrates the continuous high expression of *ANGPT1* in the longissimus dorsi muscle from 1 to 36 months post-birth. These findings imply that the *ANGPT1* gene might play a pivotal role in muscle development. The variation in the copy number of *ANGPT1* can be detected in the qPCR cycle via a specific threshold number of repetitions, which is determined based on the outcome of the control gene. The copy number spectrum typically spans from two to four copies in CNV-1, one to four copies in CNV-2, and one to eight copies in CNV-3 (Appendix A). Notably, CNV-1 does not exhibit loss type (Figure 3C). The predominant form (82.25%) of copy number in CNV-3 is the normal type, while instances of copy number loss (copy < 2) are less common in CNV-2 (Figure 3D) and CNV-3 (Figure 3E).

Additionally, we explored the correlation between *ANGPT1* CNVs and the levels of *ANGPT1* mRNA expression in the LZBG. In both muscle tissues, there was a significant association between *ANGPT1* gene mRNA expression and CNV-1 gaintypes (Figure 3C, *p* < 0.01). However, no notable correlation was found between *ANGPT1* expression and the copy numbers for others (Figure 3D,E).

### 3.3. Association between the ANGPT1 CNVs and Traits

The investigation into the relationship between CNVs in the *ANGPT1* gene and growth traits indicated that different forms of CNV-1 were notably associated with CC and BW (*p* < 0.05). CNV-1 gain-of-copies genotype goats exhibited superior phenotypic characteristics compared to those with loss or normal types (Table 1). In contrast, CNV-2 and CNV-3 did not show effects on the growth traits. Regarding the carcass traits and meat quality, variations of CNV-1 were significantly related to CW and CALM. CNV-1 gain-of-copies genotype goats had better phenotypic values compared to those with the normal type (Table 1). Furthermore, different types of CNV-2 were significantly correlated with BW, CW, and SS (Appendix A), while CNV-3 did not show significant differences in any of these traits.

## 4. Discussion

Goats excel at converting grass and forage into nutritious meat efficiently, cutting down on waste, and making the most of available resources [18]. The LZBG, an indigenous Chinese goat breed that has received less research attention, exhibits considerable genetic diversity but often suffers from muscle atrophy in its young, leading to slow growth and difficulties in standing [19,25]. Research has demonstrated the crucial regulatory role of the *ANGPT1* gene in muscle differentiation and regeneration [31]. Utilizing whole-genome resequencing, we have detected several CNVs within the *ANGPT1* gene. Consequently, we performed a series of analyses on three CNVs in the *ANGPT1* gene to investigate their possible effects on the economic traits of LZBGs.

At first, we scrutinized the protein’s structure and function associated with the *ANGPT1* gene. The motif and structure of the ANGPT1 protein exhibit significant conservation across sheep, cattle, and goats. In general, the stronger the function of a gene, the more conservative it is across species. It indicates the consistent inheritance patterns and potential involvement of the ANGPT1 protein in the growth and development of animals. In particular, CNV-1 shares a region with some ANGPT1 exons. This CNV structural variation involves specific exons, which could affect different functions like protein coding, splicing regulation, gene transport function, and transcription regulation. The actual effect depends on the exon’s sequence and its place in the gene regulatory system [9,32]. The sequence of the *ANGPT1* CNV-1 overlaps with the fibrinogen superfamily. Previous studies have shown that fibrinogen can directly form fibrin gel, regulating blood coagulation and muscle fiber development [33,34]. CNVs can affect gene expression by altering gene dosage and transcriptional structure. Variations in the number of copies within this extensive coding sequence suggest that having multiple copies may lead to gene degradation, irrespective of the exact copy number [35,36]. However, the existing research methodologies are inadequate for directly conducting experiments on goat ANGPT1 protein. Hence, we hypothesize that CNV-1’s main role might be in regulating *ANGPT1* gene expression by impacting its transcriptional structure.

Secondly, the population distribution was detected. These three CNVs were identified as being highly prevalent. Subsequently, we conducted an analysis of *ANGPT1* expression in the LZBG and its correlation with various CNVs. In line with prior research, *ANGPT1* demonstrated elevated expression levels in muscle tissues. Moreover, it sustained a high level of expression in postnatal goat muscle, remaining relatively constant with age. This consistent high expression is essential for ANGPT1’s involvement in angiogenesis and muscle fiber development. Additionally, the ANGPT1 protein supports the survival of skeletal and cardiac myocytes and boosts the proliferation and differentiation of satellite cells in vitro [37,38]. The analysis of CNVs and *ANGPT1* mRNA expression indicated that the gain-of-copies genotype of CNV-1 notably enhances *ANGPT1* gene expression. This effect was exclusively noted in CNV-1, with no discernible influence on *ANGPT1* gene expression found in other *ANGPT1* CNVs, potentially attributable to their presence within intronic regions.

Finally, we assessed how these CNVs affect economic traits in goats. CNV-1 normal types LZBGs had lower BW and smaller CC compared to those gain-of-copies genotype goats. Additionally, the CW and the CALM were superior in the gain-of-copies genotype compared to the normal type. This suggests that CNVs in the *ANGPT1* gene significantly influence growth traits and meat quality in LZBG goats, likely due to *ANGPT1*’s multiple regulatory effects on muscle development, as supported by previous research [8,34,37,38].

Angiogenesis, the process of forming new blood vessels, is crucial for providing oxygen and nutrients to developing tissues, including muscles. Proper regulation of angiogenesis is essential for normal muscle growth and regeneration [39]. In 2015, Mofarrahi and colleagues found that the expression of angiopoietin-1 increased during muscle regeneration in response to injury, suggesting its involvement in the process [8]. More recently, in 2022, Kyei discovered that knocking down *ANGPT1* inhibited the differentiation of goat skeletal muscle satellite cells. Downregulating *ANGPT1* mRNA through miR-27a-3p in SMSCs can inhibit myoblast differentiation [40]. These studies provide some insights into the potential involvement of the *ANGPT1* gene and its protein product angiopoietin-1 in muscle development. However, further research is needed to fully elucidate the specific mechanisms and functions of *ANGPT1* in muscle growth and regeneration.

In summary, we postulated that copy number variations in *ANGPT1* affect the growth attributes and meat quality of the LZBG breed by modulating *ANGPT1* expression levels in muscle tissues. It is undeniable that the overall number of samples and the use of a single goat breed may impose certain limitations on the study’s conclusions. Going forward, we aim to further investigate the molecular regulatory mechanisms underlying the impact of *ANGPT1* genetic variations on meat quality traits in our forthcoming research endeavors.

## 5. Conclusions

In this research, we carried out an in-depth analysis of three CNVs located within the *ANGPT1* gene. Our focus was on their distribution patterns, their effects on *ANGPT1* expression, and their association with economic traits. These findings revealed that various forms of CNV-1 affect *ANGPT1* gene expression and have a remarkable impact on meat and growth traits. It provides a potential molecular marker for advancing genetic improvements in goat breeding. Practical applications should prioritize conducting in-depth research on mechanisms and population effects.

## Figures and Tables

**Figure 1 animals-14-02682-f001:**
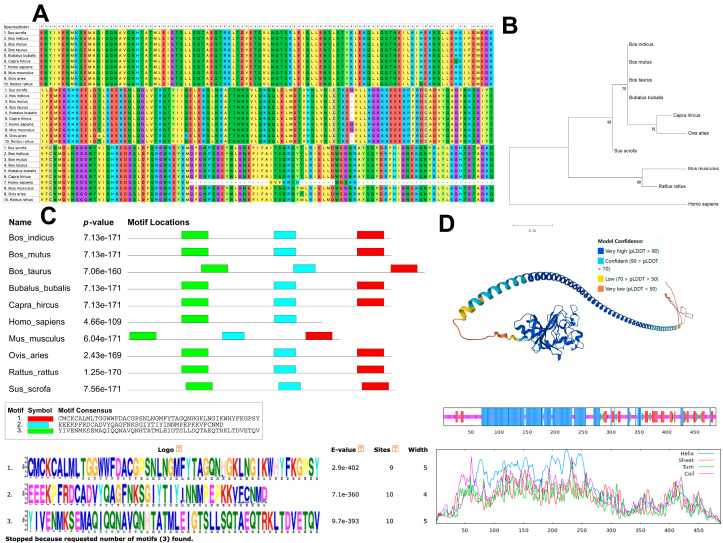
Biological evolution and conserved domains of the *ANGPT1* gene. (**A**) Multiple sequence alignment of the *ANGPT1* for 10 species. ‘*’ indicates conserved amino acid among different species. (**B**) Phylogenetic tree analysis for the *ANGPT1* gene among 10 species. (**C**) Motif structural analysis for the ANGPT1 protein among 10 species. (**D**) Protein structure prediction of the ANGPT1 protein.

**Figure 2 animals-14-02682-f002:**
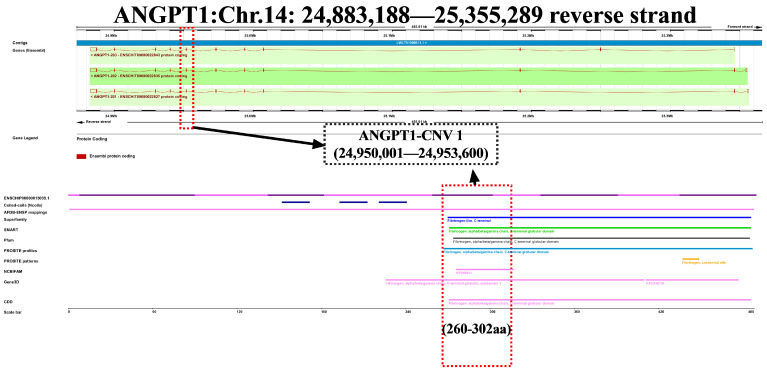
The location of the *ANGPT1*-CNV1 and the schematic diagram of overlaps between CNV-1-related protein region and conserved domains in the ANGPT1 protein sequence.

**Figure 3 animals-14-02682-f003:**
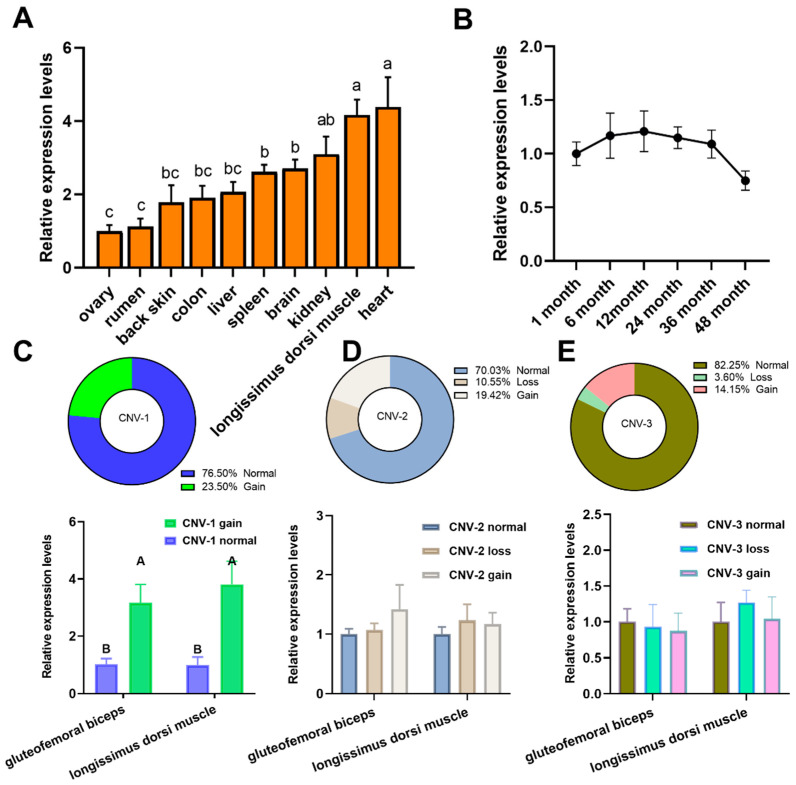
Comparison analysis of the *ANGPT1* expression levels and distribution of CNVs. (**A**) The *ANGPT1* mRNA expression profile in 10 tissues of the 12 adult-female LZBG. (**B**) Comparison of the *ANGPT1* mRNA levels at different times in the longissimus dorsi muscle of the 12 LZBG. (**C**–**E**), Distribution of CNVs different types and comparison of the *ANGPT1* expression levels among different CNVs and different genotypes in longissimus dorsi muscle and gluteofemoral triceps in the LZBG. Loss, Normal, and Gain were defined as copy number <2, =2, or ≥3, respectively. Different letters represent significant differences (a, b, c : *p* < 0.05; A, B: *p* < 0.01).

**Table 1 animals-14-02682-t001:** The association analysis between the traits and goat *ANGPT1* CNV-1.

Growth Traits	CNV Types (Mean ± SE)		*p* Values
	Normal (2 Copies) (*n* = 319)	Gain (≥3 Copies) (*n* = 98)	
body height (BH, cm)	51.57 ± 0.11	51.49 ± 0.09	0.217
body oblique length (BOL, cm)	55.87 ± 0.13	55.46 ± 0.17	0.388
chest circumference (CC, cm)	58.61 ^b^ ± 0.17	60.32 ^a^ ± 0.21	0.026
body weight (BW, kg)	20.09 ^b^ ± 0.09	21.47 ^a^ ± 0.15	0.019
cannon circumference (CAC, cm)	7.18 ± 0.02	7.17 ± 0.04	0.687
Carcass Traits and Meat Quality	Normal (2 copies) (*n* = 64)	Gain (≥3 copies) (*n* = 16)	*p* Values
carcass weight (CW, kg)	9.87 ^b^ ± 0.16	10.92 ^a^ ± 0.10	0.022
cross-sectional area of longissimus dorsi muscle(CAM, cm^2^)	7.59 ^b^ ± 0.22	8.43 ^a^ ± 0.17	0.031
water loss rate (WLR, %)	4.85 ± 0.07	4.81 ± 0.06	0.762
water holding capacity (WHC, %)	4.77 ± 0.06	4.82 ± 0.11	0.530
shear stress (SS, N)	46.83 ± 0.19	47.17 ± 0.21	0.086

Note: Different letters represent significant differences (a, b, mans *p* < 0.05).

## Data Availability

The original data presented in the study are openly available in ResearchGate at 10.13140/RG.2.2.21357.60642.

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
