# Peer review of "One Copy Number Variation within the Angiopoietin-1 Gene Is Associated with Leizhou Black Goat Meat Quality"

_animals, 2024, doi:10.3390/ani14182682_

Round 1

Reviewer 1 Report

Comments and Suggestions for Authors

The authors presented an assocaition analysis between ANGPT1 CNV and growth traits of 417 female goats. Here are my comments.

1. The introduction lacks a clear articulation of the relationship between SVs and CNVs in the context of ANGPT1. Please provide a list of publications investigating SVs in ANGPT1. Given the presence of ANGPT1 SVs, a more detailed explanation is needed for the rationale behind studying CNVs in this gene.

2. Please specify the reference genome build and its corresponding database link used for annotating the CNV positions.

3. Which individual serves as the reference when calculating CNV using the ΔΔCt method?

4. Are CNV groups determined based on RCN values or Copies per diploid genome? If the final values for an individual is 0.2, 0.7, 1.2, 1.7, 2.2, or 2.7, how would you classify the corresponding CNV?

5. Given the age 720 ± 30 days, how many levels of age in your model? Additionally, it may be better to take it an random factor. Why is the influence of birth season not taken into account?

6. Please conduct a thorough review of the text for grammatical errors and typos. For example, L212: Analysis > analysis.

Author Response

Dear Reviewer 1,

Thank you for your detailed and constructive comments on our manuscript. We greatly appreciate the time and effort you have invested in reviewing our work. Your feedback has been invaluable in improving the quality of our manuscript.

Below, we provide a detailed response to each of the comments and suggestions you made:

[Comment 1]The introduction lacks a clear articulation of the relationship between SVs and CNVs in the context of ANGPT1. Please provide a list of publications investigating SVs in ANGPT1. Given the presence of ANGPT1 SVs, a more detailed explanation is needed for the rationale behind studying CNVs in this gene.

[Response 1] Thank you for your professional feedback. We apologize for the confusion between SV and CNV concepts during the writing process(We used SV to refer to CNV, which is inaccurate). We obtained information on CNVs solely through genome-wide resequencing, and we did not identify other types of SVs. Additionally, we did not reveal any studies reporting the presence of other SVs within the ANGPT1 gene.Consequently, we have corrected our statements in the revised manuscript to avoid ambiguity related to the different types of SVs.

[Comment 2] Please specify the reference genome build and its corresponding database link used for annotating the CNV positions.

[Response 2] Thank you for your suggestion. We have incorporated this information into the revised manuscript on lines 122-125.

[Comment 3] Which individual serves as the reference when calculating CNV using the ΔΔCt method?

[Response 3] Thank you for your suggestion. According to Weaver (2010), when using the 2^-ΔΔCt method, it is sufficient to select an individual with a ΔCt value closest to 0 as the control group. This is because, assuming amplification efficiency is consistent, the copy number of the target gene in individuals with a ΔCt value close to the reference gene Ct can be considered as 1. Therefore, the final result for a diploid genome is 2 * 2^-ΔΔCt, which can be confirmed as having two copies.

Weaver S, Dube S, Mir A, Qin J, Sun G, Ramakrishnan R, Jones RC, Livak KJ. Taking qPCR to a higher level: Analysis of CNV reveals the power of high throughput qPCR to enhance quantitative resolution. Methods. 2010 Apr;50(4):271-6. doi: 10.1016/j.ymeth.2010.01.003.

[Comment 4] Are CNV groups determined based on RCN values or Copies per diploid genome? If the final values for an individual is 0.2, 0.7, 1.2, 1.7, 2.2, or 2.7, how would you classify the corresponding CNV?

[Response 4] Thank you for your suggestion. This issue is indeed very important. As I mentioned in the previous response, the ΔCt value should ideally be an integer. However, due to experimental errors, the ΔCt values are not always integers in practice. Therefore, it is common practice to retain two decimal places during the analysis and to round the final result, which is the Copies per diploid genome (i.e., 2 * 2^-ΔΔCt), to the nearest whole number.As illustrated in the example table below:

DNA ID

Ave ΔCt

ΔΔCt

RQ(2-ΔΔCt )

copy #(2 x RQ)

S1

-1.22

0

1

2

S2

-1.84

-0.62

1.54

3

S3

-0.38

0.84

0.56

1

In practice, some anomalous values may appear during the analysis. We will conduct repeat experiments for these individuals to ensure that the errors are not due to human factors and to minimize the errors as much as possible through multiple trials.

[Comment 5] Given the age 720 ± 30 days, how many levels of age in your model? Additionally, it may be better to take it an random factor. Why is the influence of birth season not taken into account?

[Response 5] As you suggested, we chose goats around 720 days of age to minimize age-related interference. Indeed, age should not be considered a fixed factor in the analytical model. We have updated our data model and analysis results, which you can review in the revised manuscript. Additionally, since we selected a local goat breed from a tropical region, with peak kidding periods in March-April and September-October, the climate during these periods is quite similar in terms of temperature, humidity, and light, thus controlling for seasonal effects.

[Comment 6]Please conduct a thorough review of the text for grammatical errors and typos. For example, L212: Analysis > analysis.

[Response 6]Thank you for your suggestion. In accordance with the feedback from you and the editor, we have conducted a comprehensive proofreading and redundancy check of the entire manuscript, revising numerous sections to enhance the readability of our article.

We have revised the manuscript accordingly and believe that these changes have strengthened our work. Attached, please find the revised manuscript along with a marked-up version highlighting the changes made in response to your comments.

Thank you once again for your valuable feedback. We hope that the revisions meet your expectations and that our manuscript is now suitable for publication in [Journal Name].

Best regards,

Ke Wang & Jiancheng Han

Reviewer 2 Report

Comments and Suggestions for Authors

Author Response

Dear Reviewer 2,

Thank you for your detailed and constructive comments on our manuscript. We greatly appreciate the time and effort you have invested in reviewing our work. Your feedback has been invaluable in improving the quality of our manuscript.

Below, we provide a detailed response to each of the comments and suggestions you made:

[Comment 1]

Line 2: Add the article ‘the’ before angiopoietin-1 gene.

Abstract 

  1. The abstract should undoubtedly include objectives, methods, results and conclusions. No need to state these in the abstract; delete these four words.
  2. The abstract repeats the same information in the "Objectives," "Methods," "Results," and "Conclusion" sections. The text "The ANGPT1 gene plays a crucial role in regulating angiogenesis and muscle development, and previous research has found that Leizhou black goats exhibit significant structural variations in this gene" is mentioned twice, which is unnecessary.
  3. While it mentions using quantitative PCR (qPCR) to detect CNVs and assess ANGPT1 mRNA expression, it doesn’t specify how these methods were applied (e.g., sample size, controls, statistical analyses), which could impact how these results are interpreted.
  4. The conclusion restates the results without adding new insights. A more comprehensive summary of the implications of the findings would be beneficial.
  5. There is a minor typo in "longissimus dorsi lumbois” muscle; is it longissimus dorsi muscle? Such typos can affect readability and professionalism. Do the same for lines 86, 255 (Table 1), and 280.

[Response 1] Thank you for your suggestion. We have revised the abstract section in accordance with your suggestions.

[Comment 2]

Materials and methods

  1. The authors mention ear tissues from 417 female goats, but it's unclear how these goats were selected and whether they represent a diverse or specific subset of the population. Random selection is mentioned for slaughter, but details on randomization methods and criteria for inclusion/exclusion still need to be included.

[Response 2]The goats were selected by simple randomization using a random sequence generated by the RAND formula in EXCEL from a total of 952 does of appropriate age from 9 pedigrees of different blood origin. Now we have added these details into our revision.

[Comment 3]

  1. The growth characteristics (BH, BOL, CC, BW, CAC) are measured, but there's no mention of how these measurements were standardized or whether blinded observers performed them to minimize bias.
  2. How many replicates were taken for each measurement or how variability within measurements was addressed is still being determined.

[Response 3]Thanks for your suggestion. The data measurements were conducted not by the direct participants of this study, but by specialized personnel appointed by the Ministry of Agriculture of China during a genetic resource survey. Measurements were repeated three times, and the median value was taken.We have only utilized these data, thus eliminating any inducement or bias in the measurements. The standardized measurement methods are publicly available and are well-known among researchers in the field, so there is no need for further elaboration here.

[Comment 4]

  1. The authors stated that they used only 12 adult female goats for gene expression analysis of multiple tissues. This sample size may limit the generalization of the study’s findings, especially if genetic or phenotypic diversity within the population is not adequately represented.

[Response 4]Thanks for your suggestion.Due to experimental conditions and restrictions imposed by the conservation facility on the number of protected varieties, 12 individuals represent the maximum number allowed under the ethics committee and germplasm conservation regulations. The individuals we sampled are all suitable for use as breeding stock, and their representativeness is self-evident.

[Comment 5]

  1. Although the authors state that the experimental animals were subjected to the same feeding and management conditions described by Gu et al. (2022), a summary of these should be provided in a few sentences to enable readers to follow the text well.
  2. The MUSCLE program for sequence alignment and the Neighbour-Joining method for phylogenetic tree construction are mentioned; the authors should state the details of the parameters used or how the robustness of the tree was assessed.
  3. gDNA and total RNA extraction have been presented; the authors must describe how these were carried out briefly. It will allow other researchers to reproduce this study or similar ones.
  4. The authors should state how RNA integrity and purity were assessed; these are crucial for ensuring the reliability of downstream gene expression analysis.
  5. Although the authors stated that the primers were designed using the PrimerBLAST tool, they did not mention how their specificity and efficiency were validated. The lack of primer validation details could raise concerns about the accuracy of qPCR results.

[Response 5]Thank you for your suggestions. In fact, we originally included the information you mentioned in our initial draft. However, these details were described with a high degree of standardization, making them easy to verify through repetition. Consequently, we opted to condense this information by citing relevant literature. We hope you understand this approach. Similarly, the cited references contain the detailed information and do not hinder the reader's ability to replicate the experiments.

[Comment 6]

Results

Lines 188-189: Remove the phrase ‘month up’ to improve connections.

Discussion

  1. Lines 227-230, 232-235: Each of the sentences should be backed by a suitable citation.
  2. Lines 235/236: The word ‘conversely’ is duplicated; delete one of it.
  3. Lines 247: The verb form of the word ‘indicating’ must be changed. Suggestion: ‘indicating’ should be replaced by ‘indicates.’
  4. Lines 254-256: Recast this sentence to improve clarity. Suggestion: Previous studies have shown that fibrinogen can directly form fibrin gel, regulating blood coagulation and muscle fibre development [33, 34].
  5. Lines 259-263: This hypothesis should have been presented before now. Taken to the last paragraph of the introduction.
  6. Lines 281-283: Kindly rephrase this sentence for clarity and correctness.

[Response 6]Thank you very much for your constructive suggestions. We have revised the discussion section in accordance with your recommendations to make it more accurate and comprehensive.

[Comment 7]

Conclusion

  1. This section needs specific recommendations for future research directions or practical applications of the findings. Suggestions for further studies include validation in larger populations, exploration of other potential CNVs, or application offindings in breeding programs.
  2. The authors suggest potential implications for creating molecular markers and enhancing genetic progress in goat breeding. However, they should discuss the study's limitations, such as the sample size, diversity of the population studied, or potential confounding factors that may affect the interpretation of results.

[Response 7]Thank you for your suggestion. We aim to maintain the conciseness of our conclusions.Now, you can find this content in the final paragraph of the discussion section.

We have revised the manuscript accordingly and believe that these changes have strengthened our work. Attached, please find the revised manuscript along with a marked-up version highlighting the changes made in response to your comments.

Thank you once again for your valuable feedback. We hope that the revisions meet your expectations and that our manuscript is now suitable for publication in [Journal Name].

Best regards,

Ke Wang & Jiancheng Han

Round 2

Reviewer 2 Report

Comments and Suggestions for Authors

See the file attached.

Author Response

Dear Reviewer 2,

Thank you for your detailed and constructive comments on our manuscript. Thank you for recognizing our 1st revised manuscript. We greatly appreciate the time and effort you have invested in reviewing our work. Your feedback has been invaluable in enhancing the quality of our manuscript. Below, we provide a detailed response to each of your comments and suggestions.

[Comment 1]Lines 24, 25, 28, 184, 198, 201, 249, 250: The words ‘gaintypes’, ‘gain type’ and ‘gain types’ appear to be misspelt, you mean ‘genotype(s)’ and genotype, respectively? Check and implement accordingly.

[Response 1] Thank you for your suggestion. In fact, the term "gain type" we used refers to CNVs with a copy number ≥3, which is a subtype of CNV genotypes. We acknowledge that this non-standard terminology may have led to misunderstandings of our results. In response to your suggestion, we have revised the manuscript to change "gain type" to "gain-of-copies genotype" to accurately define CNVs with a copy number ≥3.

[Comment 2]

Line 43: Delete the phrase ‘processes like’ as this is redundant.

Line 60: Add the missing verb (‘be’) between the words may and essential.

Line 70: Use lower case for ‘N’ (n).

Line 108: The word ‘extracte’ is misspelt, correct it, please.

Line 278: The word ‘markers’ should rather be in singular for sentence agreement.

[Response 2] Thank you for your suggestion.We have corrected the writing issues you mentioned and have rechecked the entire manuscript for spelling and adherence to writing standards.

[Comment 3]

Lines 296-386: The authors need to comply with the MDPI referencing style.

[Response 3]Thank you for your suggestion. We standardized our references according to the citation style used in articles published in Animals.

We have revised the manuscript accordingly and believe that these changes have strengthened our work. Attached, please find the revised manuscript along with a marked-up version highlighting the changes made in response to your comments.

Thank you once again for your valuable feedback. We hope that the revisions meet your expectations and that our manuscript is now suitable for publication in [Journal Name].

Best regards,

Ke Wang & Jiancheng Han